# A High Adherence to Six Food Targets of the Mediterranean Diet in the Late First Trimester is Associated with a Reduction in the Risk of Materno-Foetal Outcomes: The St. Carlos Gestational Diabetes Mellitus Prevention Study

**DOI:** 10.3390/nu11010066

**Published:** 2018-12-31

**Authors:** Carla Assaf-Balut, Nuria García de la Torre, Manuel Fuentes, Alejandra Durán, Elena Bordiú, Laura del Valle, Johanna Valerio, Inés Jiménez, Miguel Angel Herraiz, Nuria Izquierdo, María José Torrejón, María Paz de Miguel, Ana Barabash, Martín Cuesta, Miguel Angel Rubio, Alfonso Luis Calle-Pascual

**Affiliations:** 1Endocrinology and Nutrition Department, Hospital Clínico San Carlos and Instituto de Investigación Sanitaria del Hospital Clínico San Carlos (IdISSC), 28040 Madrid, Spain; carlaassafbalut90@hotmail.co.uk (C.A.-B.); nurialobo@hotmail.com (N.G.d.l.T.); aduranrh@hotmail.com (A.D.); elena.bordiu@salud.madrid.org (E.B.); lauradel_valle@hotmail.com (L.d.V.); valeriojohanna@gmail.com (J.V.); i.jimenez.varas@gmail.com (I.J.); pazdemiguel@telefonica.net (M.P.d.M.); ana.barabash@gmail.com (A.B.); cuestamartintutor@gmail.com (M.C.); marubioh@gmail.com (M.A.R.); 2Faculty of Medicine, Universidad Complutense de Madrid, 28040 Madrid, Spain; maherraizm@gmail.com (M.A.H.); nuriaizquierdo4@gmail.com (N.I.); 3Preventive Medicine Department, Hospital Clínico Universitario San Carlos and Instituto de Investigación Sanitaria del Hospital Clínico San Carlos (IdISSC), 28040 Madrid, Spain; mfuentesferrer@gmail.com; 4Gynecology and Obstetrics Department, Hospital Clínico Universitario San Carlos and Instituto de Investigación Sanitaria del Hospital Clínico San Carlos (IdISSC), 28040 Madrid, Spain; 5Clinical Laboratory Department, Hospital Clínico Universitario San Carlos and Instituto de Investigación Sanitaria del Hospital Clínico San Carlos (IdISSC), 28040 Madrid, Spain; mjosetorrejon@gmail.com; 6Centro de Investigación Biomédica en Red de Diabetes y Enfermedades Metabólicas Asociadas (CIBERDEM), 28040 Madrid, Spain

**Keywords:** pregnancy, nutrition, MedDiet, dietary patterns, gestational diabetes, maternofoetal outcomes

## Abstract

A prenatal diet affects materno-foetal outcomes. This is a post hoc analysis of the St. Carlos gestational diabetes mellitus (GDM) Prevention Study. It aims to evaluate the effect of a late first-trimester (>12 gestational weeks) degree of adherence to a MedDiet pattern—based on six food targets—on a composite of materno-foetal outcomes (CMFCs). The CMFCs were defined as having emergency C-section, perineal trauma, pregnancy-induced hypertension and preeclampsia, prematurity, large-for-gestational-age, and/or small-for-gestational-age. A total of 874 women were stratified into three groups according to late first-trimester compliance with six food targets: >12 servings/week of vegetables, >12 servings/week of fruits, <2 servings/week of juice, >3 servings/week of nuts, >6 days/week consumption of extra virgin olive oil (EVOO), and ≥40 mL/day of EVOO. High adherence was defined as complying with 5–6 targets; moderate adherence 2–4 targets; low adherence 0–1 targets. There was a linear association between high, moderate, and low adherence, and a lower risk of GDM, CMFCs, urinary tract infections (UTI), prematurity, and small-for-gestational-age (SGA) newborns (all *p* < 0.05). The odds ratios (95% CI) for GDM and CMFCs in women with a high adherence were 0.35((0.18–0.67), *p* = 0.002) and 0.23((0.11–0.48), *p* < 0.001), respectively. Late first-trimester high adherence to the predefined six food targets is associated with a reduction in the risk of GDM, CMFCs, UTI, prematurity, and SGA new-borns.

## 1. Introduction

The prenatal period is a critical moment in the development of diseases in mothers and their offspring. Modifiable risk factors such as maternal diet can modulate different functions in the expression of genes, hormone concentrations, and risk of diseases later in life.

Certain dietary patterns adopted prior to and during early pregnancy have been associated with complications in the mother and her offspring. These complications include gestational diabetes mellitus (GDM), pregnancy-induced hypertension, preeclampsia, caesarean sections, prematurity, and neonatal size [1,2,3,4,5,6,7,8,9,10].

Previously our group identified four early-pregnancy dietary patterns that are associated with a reduced risk of developing GDM, diagnosed according to the International Association of Diabetes and the Pregnancy Study Groups criteria (IADPSGc) [11]. A recent randomized controlled trial analyzed by the intention-to-treat approach, found that the adherence to a Mediterranean Diet (MedDiet), supplemented with extra virgin olive oil (EVOO) and pistachios, was associated with a 30% reduction in the incidence of GDM [12]. In addition, better materno-foetal outcomes were observed. The MedDiet seems like an optimal diet to adopt during pregnancy, in order to promote materno-foetal health.

Numerous observational studies have evaluated the effect of adopting specific dietary patterns before and during early pregnancy on several maternal, pregnancy, and neonatal complications [1,2,3,4,5,6,7,8,9,10]. A high intake of vegetables, fruits, nuts, and legumes has been associated with a lower risk of prematurity, GDM, gestational hypertension, and preeclampsia, and with high and low infant birth size [1,3,9,13]. Meanwhile, a diet rich in refined cereals, red meat, pastries, and saturated fats, and low in vegetables and fruits, has been associated with higher risk of GDM, prematurity, lower birth weight, and hypertensive disorders [1,13]. Moreover, a recent systematic review and meta-analysis suggested that a healthy diet consisting of a high intake of vegetables, fruits, legumes, whole grains, and fish improves adverse pregnancy and birth outcomes [13]. Notwithstanding, there is no conclusive evidence on what dietary pattern to follow during pregnancy.

The Mediterranean diet adherence screener (MEDAS) questionnaire is a short and simple 14-item screener that evaluates the adherence to the MedDiet [14]. However, it is not specifically adapted to pregnancy. For instance, it positively rates the consumption of alcohol and juice, when both are misadvised in pregnancy. The intake of fresh juice should not be considered to be equivalent to the consumption of a piece of fruit. These should be analyzed as independent foods, especially since the consumption of juice/sweetened beverages have been associated with the development of GDM [2,15].

We have analyzed late first-trimester dietary habits of women that complied with the protocol of the St. Carlos GDM Prevention study. We have considered six fundamental items to evaluate women’s adherence to a MedDiet in pregnancy. Three of these elements are related to the tools that are usually used to ensure compliance to the MedDiet [16]. These are the consumption of nuts and of EVOO [3]. The three remaining are foods that are associated with GDM development [2], in addition to having been scored with slightly different criteria to those of the MEDAS.

The aim of this post hoc analysis was to assess the effect of a late first-trimester degree of adherence to a MedDiet pattern—using these six items—on materno-foetal complications.

## 2. Materials and Methods

### 2.1. Study Design

This paper presents a post hoc analysis of the St. Carlos GDM Prevention Study. A detailed description of this study has been thoroughly described elsewhere [3]. It was a randomized controlled trial that was conducted from 1 January through to 31 December 2015. The primary endpoint was to compare the effects of two different types of diet—a standard diet versus a MedDiet supplemented with EVOO and pistachios—on the incidence of GDM diagnosed at 24–28 gestational weeks (GWs) in women who were normoglycemic on their first gestational visit (8–12 GWs).

All consecutive pregnant women were enrolled at 12–14 GWs during a routinely scheduled clinic consultation (first ultrasound visit) with their healthcare provider. Participants were randomized at 12–14 GWs (visit 1) to a control or intervention group, after an evaluation of inclusion criteria (8–12 GWs), and after signing the consent form. Both groups were followed up at baseline (visit 1); GDM screening, performed at 24–28 GWs (visit 2); 36–38 GWs (visit 3); and delivery. At visit 1, 2, and 3 women were interviewed about their lifestyle, and blood and urine samples were obtained. In addition, women were weighed, and their blood pressure was measured. Obstetric records were reviewed after delivery, to obtain information about the delivery and the newborns.

The intervention and control groups received dietary guidelines, based on MedDiet principles. However, the intervention group attended a group session (one week within randomization), where they were instructed to enhance the consumption of EVOO and nuts. To facilitate compliance, women were given 10 L of EVOO, and 2 kg of pistachios at both the group session and visit 2. Meanwhile, the control group was told to limit the consumption of all types of fats (including <3 servings/week of nuts and <40 mL/day of EVOO.

The study was approved by the Ethics Committee of Hospital Clínico San Carlos, and conducted according to the Helsinki Declaration. All women signed a letter of informed consent. The study was registered at ISRCTN84389045.

For the present post hoc analysis, the variables were retrieved from the main study database. The study population was treated as a cohort, independent of randomization assignment. The sample was stratified into three groups according to dietary patterns in late first trimester (from 12–14 to 24–28 GWs). These were recorded at visit 2. The earliest opportunity to provide a nutritional intervention to pregnant women is usually from 12–14 GWs onwards. Therefore, dietary habits were evaluated after this moment.

### 2.2. Dietary Patterns Classification

For the current study, six food targets were chosen from the MedDiet. The first three were considered as tools to ensure compliance to the MedDiet. These were an intake of ≥40 mL/day of EVOO, of EVOO >6 times/week, and of >3 servings/week of nuts. The other three were elements that were scored with slightly different criteria to those used in the MEDAS questionnaire. These were >12 servings/week of vegetables (raw or cooked), >12 servings/week of whole fruits (excluding fresh juice), <2 servings/week of juice (fresh or bottled). A high adherence was set for achieving 5–6 targets; a moderate adherence for achieving 2–4 targets; and a low adherence for achieving 0–1 targets.

### 2.3. Dietary Assessments

Two different semi-quantitative questionnaires were used to evaluate lifestyle and diet during the study period: the Diabetes Nutrition and Complications Trial (DNCT), and MEDAS questionnaires. The dietitian applied these two questionnaires at visits 1, 2, and 3 in a face-to-face interview with the participant.

The DNCT questionnaire was used to evaluate physical activity and general healthy eating habits. It contains 15 items, three of which evaluate physical activity, and the 12 remaining, food frequency intakes (daily or weekly). A detailed description has been published previously [17]. This questionnaire was used to obtain the “Nutrition score”. This score was used to assess the global quality of the participants’ lifestyle, considering a score of >5 to be the objective. The food frequency intakes were also used to obtain the mean consumption of each food item independently. In this way, specific dietary patterns and compliance with pre-defined targets were evaluated.

The MEDAS questionnaire is a short screener containing 14 food items that evaluate the adherence to a MedDiet pattern. This questionnaire has been developed and validated in the PREDIMED study [14]. These 14 items mostly provide information about the consumption of vegetables and fruits, legumes, nuts, EVOO, oily fish, white and red meat, and wine. In the MEDAS questionnaire, the compliance to each food item provides +1 points. A total of >10 is considered ideal. In this study, the scoring of this questionnaire was labelled the “MEDAS score”. There are some differences with the original MEDAS criteria. We did not account for the consumption of alcohol or juice, because both are misadvised during pregnancy. Therefore, fresh juice intake was excluded from the item “fruit units” in the questionnaire, and an intake of >12 fruits/week (not >21) was considered favorable. The consumption of fruit juices (fresh or bottled) were included in the item of sweetened beverages. Also, vegetable intake was accounted for, regardless of being raw or cooked. Pregnancy guidelines indicate that vegetables should be cleaned thoroughly, if they are to be eaten raw. This sometimes leads to limiting the intake of raw vegetables during pregnancy, and preferring to eat cooked vegetables. Given the above, the ideal MEDAS score was considered to be >8. These changes were made, to be able to use a questionnaire that evaluates adherence to the MedDiet in pregnancy.

### 2.4. Study Population

There were no differences between this post hoc analysis and the original St. Carlos GDM Prevention Study, in that neither the inclusion or exclusion criteria nor the randomization were used.

A total of 1501 pregnant women were eligible, and invited to participate. One thousand women signed the consent form, and were randomized to control or intervention group. Of these, 874 women complied with the study protocol. Consequently, they were stratified intro three groups, according to the late first-trimester degree of adherence to the six food targets (Figure 1).

### 2.5. Outcomes Measures

The primary endpoint for this post hoc analysis was to evaluate the associations of late first-trimester degrees of adherence to the six food targets (high, moderate, and low) with the risk of GDM and a composite of materno-foetal complications (CMFC). The CMFC was defined as having at least one of the following: emergency C-section, perineal trauma, pregnancy-induced hypertension and preeclampsia, prematurity, large-for-gestational-age, and/or small-for-gestational-age newborns.

Secondary outcomes were to compare between the groups of different degrees of adherence, clinical and biochemical characteristics, and materno-foetal complications (analyzed individually).

### 2.6. Other Assessments

#### 2.6.1. Participant’s History

Participant’s histories were aspects such as any family history of metabolic disorders (type 2 diabetes mellitus and metabolic syndrome, when >2 components were present in the same relative), obstetric history (miscarriages and/or GDM), educational level, employment, number of prior pregnancies, smoking habits, and gestational age at entry (according to the first ultrasound). These set of data were collected at baseline.

#### 2.6.2. Anthropometric Data

Blood pressure, height, weight, gestational weight gain, and body mass index (BMI) were evaluated and recorded at all three visits. Blood pressure was measured with an electronic, digital sphygmomanometer Omron 705IT (Omron Global, Kyoto, Japan). Weight gain was calculated according to self-referred pregestational body weight. Height and weight were measured by using a digital height measuring scale Seca 769 (seca GmbH & Co. KG, Hamburg, Germany).

#### 2.6.3. Maternal, Delivery, and Neonatal Outcomes

Maternal: GDM, pregnancy induced hypertension, preeclampsia, albuminuria, and urinary tract infections (number of events requiring antibiotic treatment).

Delivery: type of delivery (vaginal, instrumental, or caesarean section) and perineal trauma.

Neonatal: birth weight and height, large-for-gestational-age (>90th percentile), small-for-gestational-age (<10th percentile), pH cord blood, Apgar score, neonatal hypoglycemia, hyperbilirubinemia, respiratory distress, and admission to a neonatal intensive care unit (NICU).

#### 2.6.4. Biochemical Analysis

Laboratory tests were scheduled for each visit. Blood was drawn after an overnight fast of 8–10 h, between 08:00 and 09:00 a.m. HbA1c, standardized by the International Federation of Clinical Chemistry and Laboratory Medicine (IFCC) was determined by using ion-exchange high-performance liquid chromatography in gradient, with a Tosoh G8 analyzer (Tosoh Co., Tokyo, Japan). Serum insulin was determined by a chemiluminescence immunoassay in an Inmmulite 2000 Xpi (Siemens, Healthcare Diagnostics, Munich, Germany). HOMA-insulin resistance (HOMA-IR) was calculated as glucose (mmol/L) × insulin (mcUI/mL)/22.7.

In an Olympus 5800 (Beckman-Coulter, Brea, CA, USA), serum levels of HDL-cholesterol and creatinine were measured using direct and Jaffé kinetic method, respectively. LDL-cholesterol was calculated with the Friedewald formula. Dimension Vista (Siemens Healthcare Diagnostics, Munich, Germany) was used to measure apolipoprotein B and C-RP by immunonephelometry and nephelometry, respectively.

Serum levels of fasting plasma glucose, total cholesterol, triglycerides, and albumin were measured by using a colorimetric method with glucose–hexokinase, CHOD-PAP, GPO-PAP, and green bromocresol, respectively.

The quality of the methods was evaluated monthly by the External Quality Assurance Program of the SEQC (Sociedad Español Química Clínica).

#### 2.6.5. Statistical Analysis

Categorical variables were expressed, with their frequency distribution and continuous variables as means and standard deviation, or means and 95% confidence interval (CI). The Saphiro–Wilk test was used to verify the normal distribution of the data. For continuous variables, the Kruskall–Wallis test or a one-way analysis of variance (ANOVA) were applied. For categorical variables, the χ^2^ test was performed. The nonparametric Wilcoxon matched-pairs test was performed for related variables.

The Chi square for linear trends and unadjusted logistic regression analysis (women with a low adherence reference group) was performed, to evaluate the relationship between the different degrees of adherence, and the binary primary and secondary outcomes. For the GDM and CMFC outcomes, a subgroup analysis was conducted, introducing the interaction term between the adherence and the stratification variables (BMI, age, and parity). Two multivariable final models were adjusted, to assess the effect of adherence of GDM and CMFC, adjusting for BMI, age, and sex. All p-values are 2-tailed at less than 0.050. Analyses were done using SPSS, version 21 (SPSS, Chicago, IL, USA).

## 3. Results

### 3.1. Demographics

Table 1 summarizes the socio-demographic characteristics of the women included in this study. The mean age was 32.9 ± 5.1 years old. The cohort was diverse in ethnicity, with a majority of Caucasian women (67.2%). Also, the majority of women were primiparous (43.3%), and had an equivalent of a university degree (66.0%). On average, prepregnancy BMI was 23.1 ± 3.8 kg/m^2^, and 23.9 ± 3.9 kg/m^2^ at baseline.

### 3.2. Dietary Habits at Baseline and 24–28 GWs

The mean (95% CI) consumptions of the different food groups at baseline, and at 24–28 GWs of all women are shown in Table 2. Overall, there were improvements in the intakes of vegetables, portions of fruit, nuts, EVOO, whole grain bread and cereals, legumes, low-fat dairy products, fortified dairy products, and homemade sauces. The Nutrition and MEDAS scores also increased.

### 3.3. Socio-Demographic, Clinical, and Biochemical Characteristics of Women According to Late First-Trimester Degree of Adherence to Six Food Targets

Table 3 shows baseline characteristics of women, stratified into groups by degree of adherence. There were differences between groups in the distributions of ethnicity, family history of metabolic disorders (type 2 diabetes and metabolic syndrome), education level, and parity. There was a positive correlation with age, Caucasian ethnicity, and education level (all *p* < 0.01).

### 3.4. Dietary Habits According to the Late First-Trimester Degrees of Adherence to Six Food Targets at Baseline and at 24–28 GWs

A total of 115 (13.1%) women complied with 5–6 targets, 623 (71.3%) with 2–4, and 136 (15.6%) with 0–1. There was a linear association between the degree of adherence to the six food targets, and the MEDAS (r = 0.760; *p* < 0.001) and Nutrition scores (r = 0.707; *p* < 0.001). Table 4 shows the mean (95% CI) consumption of different food groups at baseline and at 24–28 GWs, according to the degree of adherence.

### 3.5. Associations between Late First-Trimester Degrees of Adherence and Materno-Foetal Complications

Table 5 shows the incidence of different materno-foetal complications according to the degree of adherence. It also shows the crude logistic regression analysis of the risk of having materno-foetal complications, according to the degree of adherence. The higher the adherence to the six food targets, the lower the incidence and risk of GDM, urinary tract infections, prematurity, SGA newborns, and CMFCs (*p*-trend all < 0.01). No significant differences were found in the other analyzed variables.

Figure 2a,b show the logistic regression analysis by subgroups for the probability of having GDM and CMFC, according to the level of adherence (in crude and stratified by BMI, age and parity). An interaction was observed between the BMI and the level of adherence (high versus low and moderate versus low) for the incidence of GDM (*p* = 0.033) (Figure 2a). The protective effect of having a high adherence to the six food targets was higher in women who are overweight and obese, than those with a normal weight. After adjusting for age and parity, the results remained similar: “high adherence” (OR 0.04 (95% CI 0.005–0.342), *p* = 0.003) and “moderate adherence” [OR 0.31 (95% CI 0.15–0.64), *p* = 0.002] in overweight/obese women; “high adherence” [OR 0.65 (95% CI 0.301–1.400), *p* = 0.271] and “moderate adherence” (OR 0.69 (95% CI 0.39–1.24), *p* = 0.214) in normal weight women. For the incidence of CMFCs, there was an interaction between parity and the level of adherence (*p* = 0.046) (Figure 2b). The protective effect of having a high adherence to the six food targets was higher in multiparous than primiparous women. After adjusting for BMI and age, the results remained similar: high adherence (OR 0.13 (95% CI (0.03–0.60), *p* = 0.008), moderate adherence (OR 0.75 (95% CI 0.42–1.35), *p* = 0.336) in multiparous women; high adherence (OR 0.25 (95% CI 0.10–0.64), *p* = 0.004) and moderate adherence (OR 0.32 (95% CI 0.17–0.61), *p* = 0.001) in primiparous women.

## 4. Discussion

The current study evaluated how the compliance to six food targets of the MedDiet—between the late first-trimester and second-trimester—could be associated with the presence of materno-foetal outcomes. The analysis was performed from the late first trimester, because it is usually the earliest prenatal appointment, and it presents the earliest opportunity to provide nutritional education. The main findings indicate inverse associations between high-, moderate-, and low-adherence to the six food targets, and the incidence of materno-foetal complications. Having a high adherence is associated with a 65% reduced risk of developing GDM, and a 77% reduced risk of having CMFCs. The risk of urinary tract infections, prematurity, and SGA newborns were also reduced. To our knowledge, no other studies have evaluated the associations of different degrees of adherence to these six food targets with materno-foetal complications.

The results obtained confirm that generally, women seem to improve their dietary habits during pregnancy. This is observed both when diet is assessed as individual food groups, and as dietary patterns (reflected by the Nutrition and MEDAS scores). Pregnancy is known to be a motivating phase for women where positive lifestyle changes are implemented [18,19]. In general, these women tend to drift towards a healthier dietary pattern. In agreement with other studies, an analysis of global dietary changes revealed that women improved their weekly intake of vegetables, fruit, nuts, EVOO, wholegrain bread and cereals, legumes, and low-fat dairy products [20,21]. Moreover, women did seem to improve their weekly intake (days/week) of EVOO. However, when analyzing it in mL/day, results showed that women seem to find it to more challenging to comply with consuming > 40 mL/day.

Results show a linear trend between the degree of adherence to the six food targets and having better dietary habits in late first trimester. The higher the adherence, the higher the consumption of vegetables, fruits, nuts, EVOO (days/week), oily fish, whole-grain cereals, legumes, skimmed dairy products, EVOO (mL/day), and homemade sauces. This was in parallel to a lower consumption of refined flours, red and processed meat, juices/sweetened drinks, pastries, and biscuits. Moreover, the degree of compliance to the six food targets seemed to correlate positively with both the MEDAS and Nutrition scores. The mean MEDAS score in the high adherence group met the ideal score > 8. Therefore, this could indicate that the use of these six food-groups could be able estimate an optimal adherence to the MedDiet.

The ideal dietary pattern was defined as obtaining a high adherence to the proposed food targets (5–6 targets). Only 13.1% women achieved this. Women who belonged to this group were older, more frequently of Caucasian origin, primiparous, and had a higher educational level. In comparison to groups of lower adherence, the group with a high adherence was associated with significantly lower risks of numerous materno-foetal complications. However, women with a moderate adherence— who only had to comply with 2–4 targets—also reduced their risk of developing complications. Therefore, discrete improvements in the dietary habits are favorable, even if an ideal adherence is not achieved.

This study shows that women of younger age, with a lower educational level, and multiparous, should be targeted the most, since they tend to have worse dietary habits. This is consistent with results from other studies [22]. Moreover, an analysis of interactions revealed that special attention should be paid to women who are overweight/obese, and who are multiparous. Interactions indicate that these groups of women could benefit the most from nutritional interventions, and from improvements in dietary habits in pregnancy. In preventing GDM and CMFCs, the level of adherence to the six food targets was more relevant in these groups of women.

Different dietary patterns—both healthy and unhealthy—have been studied in association with materno-foetal health. Most of the evidence suggests that a “western diet” is deleterious, while a “healthy/prudent” pattern is beneficial [1,13,23]. Most of the “healthy” dietary patterns described in the literature include a high consumption of plant-based foods (vegetables, fruits, nuts, and legumes), vegetable oil (mostly olive oil), fish, poultry, and low-fat dairy products. This dietary pattern profile, within its variations, has been associated with a lower risk of GDM, prematurity, gestational hypertension, and preeclampsia, and high and low infant birth sizes [3,4,5,6,7,8,9,13,22,24]. As reflected by this study’s results, the compliance to the six food targets is associated with a lower risk of GDM, urinary tract infections, prematurity, SGA newborns, and CMFCs. This is in concordance with results of other studies, where following a MedDiet pattern was also associated with a reduced risk of GDM [25] and SGA newborns [26]. A recent study showed that the adherence to a MedDiet and the intake of olive oil was associated with a reduced risk of SGA newborns [27]. No significant associations were found with the prevention of hypertensive disorders and preeclampsia, as opposed to other studies [8,10]. This could be due to the low number of cases found in our studied sample. However, we did find lower risks for prematurity in contrast to other studies that did not yield these same results [1,4]. Moreover, there seems to be no evidence in the available literature indicating any relation between maternal diet and urinary tract infections. The lower rates of urinary tract infections found in this study could be explained by the consumption of EVOO. Bioactive compounds like hydroxytyrosol—eliminated in urine after EVOO consumption—might favor the urinary microbiome, and hence lower the risk of urinary tract infections. Other possible explanations have been explored regarding the influence of MedDiet components on materno-foetal outcomes [28,29,30,31,32,33].

## 5. Limitations

There are some limitations in this study. The six food targets used were based on components of the MedDiet did not include whole-wheat cereals, pulses, and fish. However, four fundamental food groups of this dietary pattern were considered: EVOO, nuts, fruits (excluding fresh and artificial juices), and vegetables. The consumption of these four food groups seems to indirectly translate into having overall healthy habits. Also, there was a positive correlation between the MEDAS score and the degree of adherence. Moreover, the original study was designed to answer a different specific question to the one proposed in this paper. However, the performance of post hoc analyses enables us to learn the most from the data that we have from the St. Carlos GDM Prevention Study.

## 6. Conclusions

This study suggests that the higher the adherence to six food targets of the MedDiet, the lower the risk of developing GDM, CMFS, urinary tract infections, prematurity and SGA newborns. Moreover, the six food targets used in this study seem appropriate to evaluate adherence to the MedDiet. This reinforces the importance of providing early nutritional education to pregnant women, where pregnancy presents a unique opportunity to promote lifestyle changes. The long-term consequences of adopting this diet profile during pregnancy on both mother and offspring are being currently studied.

## Figures and Tables

**Figure 1 nutrients-11-00066-f001:**
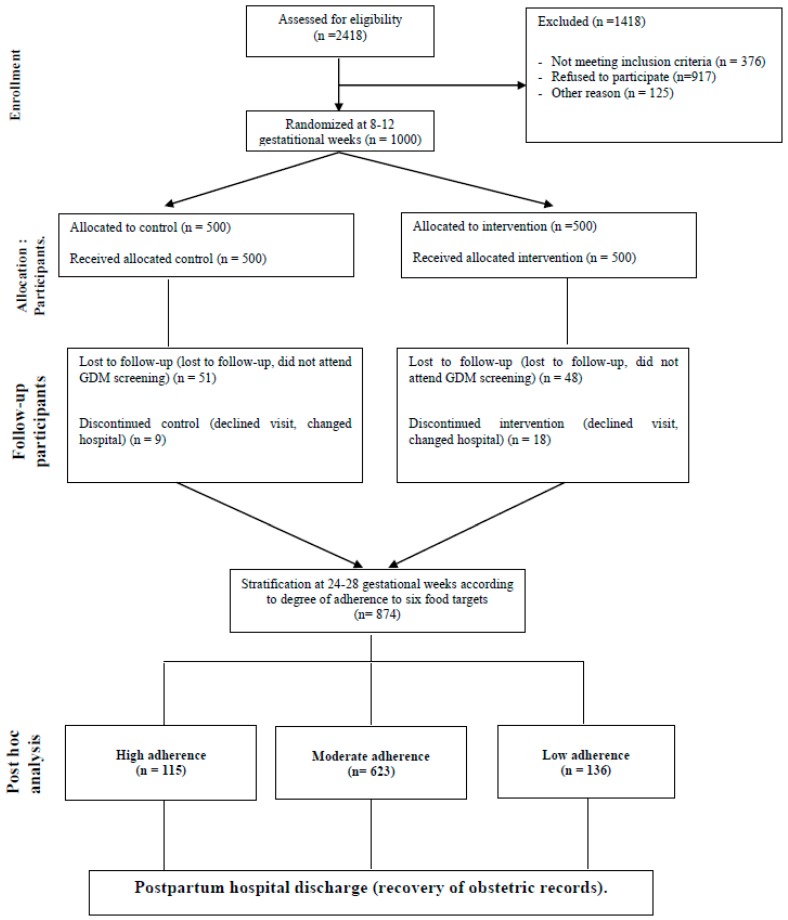
Flow diagram showing a summarized description of the study design, and the women analyzed for this post hoc analysis of the St. Carlos gestational diabetes mellitus (GDM) Prevention Study.

**Figure 2 nutrients-11-00066-f002:**
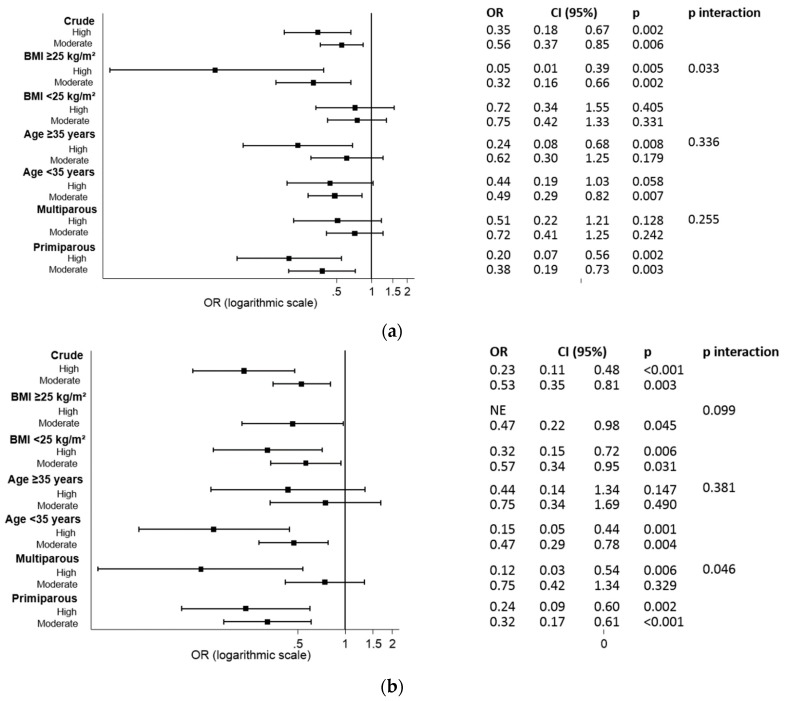
Logistic regression analysis by subgroups (crude and stratified by BMI, age, and parity) evaluating the degrees of adherence to the six food targets, and risk of (**a**) GDM; and (**b**) CMFC. *p*: logistic regression analysis comparing ORS with the reference group (low adherence group). p-interaction: logistic regression analysis evaluating the interactions between the degrees of adherence and BMI (≥25 or <25 kg/m^2^), age (≥35 or <35 years), and parity (multiparous or primiparous). OR: odds ratio; CI: confidence interval; CMFC, composite of materno-fetal complications; BMI: body mass index. High: high adherence, 5–6 targets; Moderate: moderate adherence, 2–4 targets; Low: low adherence, 0–1 targets.

**Table 1 nutrients-11-00066-t001:** Characteristics at baseline of all participants.

Variables	Total Sample (*n* = 874)
Age (years)	32.9 ± 5.1
Race/Ethnicity	
Caucasian	587 (67.2)
Hispanic	260 (29.7)
Others	27 (3.1)
Family history of	
Type 2 diabetes	180 (20.6)
Family history of MetS (>2 components)	183 (20.9)
Previous history of	
Gestational diabetes	25 (2.9)
Miscarriages	278 (31.8)
Educational status	
Elementary education	88 (8.9)
Secondary school	212 (24.3)
University degree	577 (66.0)
UNK	7 (0.8)
Employment	674 (77.1)
Number of pregnancies	
Primiparous	378 (43.3)
Second pregnancy	281 (32.2)
>2 pregnancies	215 (24.5)
Never smoker	477 (54.6)
Current smoker	72 (8.2)
Gestational age at baseline (weeks)	12.1 ± 0.6
Prepregnancy body weight (kg)	60.8 ± 10.8
Body weight at entry (kg)	62.9 ± 11.1
Weight gain (kg)	2.1 ± 3.0
Prepregnancy BMI (kg/m^2^)	23.1 ± 3.8
BMI at baseline (kg/m^2^)	23.9 ± 3.9
Systolic blood pressure (mm Hg)	107 ± 11
Diastolic blood pressure (mm Hg)	66 ± 10
Fasting blood glucose (mg/dL)	81 ± 6
HbA1c %	5.1 ± 0.3
Cholesterol mg/dL	174 ± 30
Triglycerides mg/dL	82 ± 40
TSH mcUI/mL	2.0 ± 1.3
T4L ng/dL	8.6 ± 1.5
MEDAS score	4.9 ± 1.7
Nutrition score	0.4 ± 3.2

Data are Mean ± SD or *n* (%). MetS, Metabolic Syndrome. UNK, unknown. BMI, body mass index; MEDAS Score: Mediterranean Diet Adherence Screener Score.

**Table 2 nutrients-11-00066-t002:** Consumption of individual food groups at baseline (12–14 GWs), and at 24–28 GWs.

	Total Sample				
	Servings/Week	At Targets	Servings/Week	At Targets	
Variables	Baseline	*n* (%)	24–28 GWs	*n* (%)	*p* Value
Vegetables	6.3 (6.0–6.3)	146 (16.7)	7.1 (6.8–7.4)	179 (20.1)	0.001
Pieces of fruit	10.7 (10.1–11.2)	420 (40.1)	15.0 (14.5–15.5)	643 (72.8)	0.001
Nuts	1.4 (1.3–1.5)	122 (14.0)	2.6 (2.5–2.9)	305 (32.9)	0.001
EVOO (days/week)	6.1 (6.0–6.3)	738 (84.4)	6.4 (6.3–6.5)	750 (85.8)	0.001
Oily fish or ibérico ham	1.0 (1.0–1.2)	9 (1.0)	1.0 (0.9–1.2)	66 (7.6)	0.264
Canned fish	1.5 (1.4–1.6)	20 (2.3)	1.2 (1.1–1.3)	64 (7.2)	0.001
White fish	1.2 (1.1–1.3)	N.A.	1.2 (1.1–1.3)	N.A.	0.499
Shellfish	0.2 (0.2–0.3)	N.A.	0.2 (0.1–0.2)	N.A.	0.026
Whole grain bread and cereals	1.9 (1.7–2.1)	157 (18.0)	2.5 (2.3–2.7)	215 (24.6)	0.001
White rice, bread and/or pasta	5.2 (5.0–5.3)	77 (8.7)	4.8 (4.7–5.0)	83 (9.5)	0.001
Legumes	1.7 (1.6–1.8)	167 (19.1)	1.8 (1.7–1.9)	198 (22.6)	0.005
Skimmed dairy products	2.6 (2.4–2.9)	10 (1.1)	3.3 (3.1–3.5)	357 (40.8)	0.001
Semi-skimmed dairy products	2.7 (2.5–3.0)	N.A.	2.8 (2.5–3.0)	N.A.	0.982
Full-fat dairy products	4.4 (4.2–4.6)	N.A.	4.4 (4.2–4.6)	N.A.	0.780
Supplemented dairy products	1.3 (1.2–1.5)	N.A.	2.0 (1.8–2.2)	N.A.	0.001
Butter	1.6 (1.4–1.7)	N.A.	1.4 (1.2–1–5)	N.A.	0.003
Red meat	2.3 (2.2–2.4)	282 (32.3)	1.9 (1.9–2.0)	392 (46.2)	0.001
Low-fat processed cold meat	2.7 (2.6–2.9)	204 (23.3)	2.3 (2.2–2.5)	297 (35.0)	0.001
Processed red meat	0.8 (0.7–0.9)	455 (52.1)	0.5 (0.4–0.5)	563 (64.4)	0.001
Poultry, turkey or rabbit	3.0 (2.9–3.1)	N.A.	2.8 (2.7–2.9)	N.A.	0.001
Commercial sauces	0.8 (0.7–0.9)	553 (63.3)	0.4 (0.3–0.4)	650 (74.4)	0.001
Juices and/or sweetened drinks	3.7 (3.5–3.9)	283 (32.4)	2.9 (2.7–3.1)	364 (42.9)	0.001
Pastries and biscuits	4.7 (4.5–4.9)	136 (14.4)	4.1 (3.9–4.3)	179 (21.1)	0.001
Coffees	6.9 (6.5–7.4)	N.A.	4.2 (3.9–4.5)	N.A.	0.001
Homemade sauces	2.3 (2.1–2.4)	48 (5.5)	2.4 (2.3–2.6)	50 (5.9)	0.021
EVOO mL/day	28 (26–29)	55 (17.8)	29 (28–31)	211 (24.1)	0.176
Nutrition score	0.4 (0.1–0.6)	N.A.	2.6 (2.4–2.9)	N.A.	0.001
MEDAS score	4.9 (4.8–5.0)	N.A.	6.7 (6.5–6.8)	N.A.	0.001
Physical activity Score ≥ 0	99 (11.3)	N.A.	58 (6.9)	N.A.	0.406

Results are expressed as means (95% CI) or *n* (%). *p*-value, differences of mean food intakes at baseline and at 24–28 GWs, analyzed with the non-parametric Wilcoxon matched-pairs test. GW, gestational weeks. N.A.: not applicable. EVOO: extra-virgin olive oil; MEDAS, Mediterranean Diet Adherence Screener Score. Physical Activity Score ≥0: a. Walking daily (>5 days/week). Score 0: At least 30 min; Score +1, if >60 min; Score −1, if <30 min. b. Climbing stairs (floors⁄day, >5 days a week): Score 0, Between 4 and 16; Score +1, >16; Score −1: <4). Targets: vegetables >12 servings/week); fruits: >12 servings/week; nuts > 3 servings/week; EVOO > 6 servings/week; oily fish > 3 servings/week; canned fish; wholegrain bread and cereals > 6 servings/week; white rice, bread and/or pasta < 1 serving/week; legumes > 2 servings/week; skimmed dairy products > 6 servings/week; red meat < 2 servings/week; low-fat cold meats < 2 servings/week; processed red meats < 2 servings/week; commercial sauces < 2 servings/week; juices and/or sweetened drinks < 2/week; pastries/biscuits < 2 servings/week; homemade sauces, <2 servings/week commercial sauces and >5 servings/week of sofrito; EVOO mL/day, 40 mL.

**Table 3 nutrients-11-00066-t003:** Characteristics of the clinical trial population at baseline according to late first-trimester degrees of adherence to the six food targets.

Variables	Groups			*p* Value
	Low Adherence (*n* = 136/15.6%)	Moderate Adherence (*n* = 623/71.3%)	High Adherence (*n* = 115/13.1%)	
Age (years)	31.2 ± 6.0	33.1 ± 4.9	33.9 ± 4.9	0.001
Race/Ethnicity				
Caucasian	79 (58.1)	420 (67.4)	88 (76.5)	0.005
Hispanic	51 (37.5)	188 (30.2)	21 (18.3)	
Others	6 (4.4)	15 (2.4)	6 (5.2)	
Family history of:				
Type 2 diabetes	28 (20.6)	131 (21.0)	21 (18.3)	0.019
MetS (>2 components)	18 (13.2)	135 (21.7)	30 (26.1)	
Previous history of:				
Gestational diabetes	7 (5.1)	15 (2.4)	3 (2.6)	0.342
Miscarriages	45 (33.1)	196 (31.5)	37 (32.2)	
Educational status				
Elementary education	19 (14.0)	54 (8.7)	5 (4.3)	0.001
Secondary school	49 (36.0)	145 (23.3)	18 (15.7)	
University degree	65 (47.8)	420 (67.4)	92 (80.0)	
UNK	3 (2.2)	4 (0.6)	0 (0)	
Employment	100 (73.5)	482 (77.4)	92 (80.0)	0.815
Number of pregnancies				
Primiparous	50 (36.8)	274 (44.0)	54 (47.8)	0.017
Second pregnancy	43 (31.6)	196 (31.5)	42 (36.8)	
>2 pregnancies	43 (31.6)	153 (24.5)	19 (15.4)	
Smoker				
Never	75 (55.1)	334 (53.6)	68 (59.1)	0.260
Current	16 (11.8)	52 (8.3)	4 (3.5)	
Gestational age (weeks) at baseline	12.1 ± 0.7	12.1 ± 0.5	12.0 ± 0.5	0.137
Body Weight (kg)				
Prepregnancy	61.1 ± 11.8	61.0 ± 10.8	59.5 ± 9.3	0.383
At entry	63.3 ± 11.3	63.1 ± 11.1	61.5 ± 8.4	0.356
Weight gain	2.2 ± 3.2	2.0 ± 3.0	2.0 ± 2.7	0.707
BMI (kg/m^2^)				
Prepregnancy	23.4 ± 4.1	23.2 ± 3.8	22.5 ± 3.4	0.137
At baseline	24.3 ± 4.3	23.9 ± 3.9	23.3 ± 3.5	0.111
Blood pressure (mm Hg):				
Systolic	107 ± 10	107 ± 11	107 ± 10	0.972
Diastolic	66 ± 15	66 ± 9	66 ± 8	0.809
Fasting blood glucose mg/dL	82 ± 5	81 ± 6	81 ± 7	0.377
HbA1c %	5.2 ± 0.2	5.2 ± 0.3	5.1 ± 0.3	0.436
Cholesterol mg/dL	171 ± 28	175 ± 31	176 ± 25	0.507
Triglycerides mg/dL	82 ± 38	83 ± 42	76 ± 30	0.288
TSH mcUI/mL	1.9 ± 1.2	2.0 ± 1.3	2.1 ± 1.4	0.610
T4L ng/dL	8.6 ± 1.3	8.6 ± 1.6	8.9 ± 1.2	0.144

Data are Mean ± SD or *n* (%). P Differences between groups analyzed with ANOVA (continuous variables) and χ^2^ test (categorical variables). MetS, Metabolic Syndrome. UNK, unknown. BMI, body mass index.

**Table 4 nutrients-11-00066-t004:** Consumption of individual food groups (servings/week) at baseline (12–14 GWs) and at 24–28 GWs, according to the late first-trimester degrees of adherence to the six food targets.

Variables	Groups							
	Low Adherence (*n* = 136/15.6%)		Moderate Adherence (*n* = 623/71.3%)		High Adherence (*n* = 115/13.1%)			
	Baseline	24–28 GW	Baseline	24–28 GW	Baseline	24–28 GW	*p* *^a^*	*p* *^b^*
Vegetables	4.7 (4.1–5.2)	4.2 (3.8–4.7)	6.3 (6.0–6.7)	6.8 (6.5–7.2)	8.2 (7.4–9.1)	12.2 (11.6–12.9)	0.001	0.001
Pieces of fruit	8.0 (6.6–9.4)	8.5 (7.2–9.7)	10.6 (10.0–11.3)	15.7 (15.1–16.3)	13.9 (12.4–15.4)	18.8 (17.7–19.9)	0.001	0.001
Nuts	0.9 (0.7–1.0)	0.7 (0.5–0.9)	1.4 (1.2–1.5)	2.6 (2.4–2.8)	2.2 (1.7–2.6)	5.5 (5.0–5.9)	0.001	0.001
EVOO (days/week)	4.8 (4.3–5.3)	4.2 (3.7–4.8)	6.4 (6.2–6.5)	6.8 (6.7–6.9)	6.5 (6.2–6.8)	7.0 (6.9–7.0)	0.001	0.001
Oily fish or Iberico ham	0.8 (0.7–1.2)	0.8 (0.7–1.0)	1.0 (1.0–1.1)	1.0 (0.7–1.1)	1.1 (1.0–1.3)	1.2 (1.0–1.4)	0.035	0.025
Canned fish	1.6 (1.3–1.9)	1.0 (0.8–1.2)	1.5 (1.4–1.6)	1.2 (1.1–1.3)	1.4 (1.1–1.6)	1.2 (1.0–1.5)	0.001	0.293
White fish	1.1 (0.9–1.3)	1.0 (0.8–1.2)	1.2 (1.1–1.3)	1.2 (1.1–1.3)	1.5 (1.2–1.7)	1.4 (1.2–1.7)	0.345	0.009
Shellfish	1 (0–2)	0.1 (0.0–0.2)	0 (0–0)	0.2 (0.2–0.2)	0.3 (0.2–0.4)	0.2 (0.1–0.3)	0.287	0.380
Whole grain bread and cereals	1.8 (1.3–2.2)	1.8 (1.4–2.3)	1.9 (1.7–2.1)	2.5 (2.2–2.7)	2.2 (1.6–2.7)	3.4 (2.8–4.1)	0.504	0.001
White rice, bread and/or pasta	5.5 (5.1–5.8)	5.4 (5.1–5.8)	5.2 (5.0–5.3)	4.9 (4.7–5.1)	5.0 (4.5–5.4)	3.9 (3.3–4.4)	0.217	0.001
Legumes	2.0 (1.7–2.2)	1.9 (1.6–2.2)	1.7 (1.6–1.7)	1.8 (1.7–1.9)	1.6 (1.4–1.7)	1.8 (1.6–2.1)	0.010	0.593
Skimmed dairy products	1.7 (1.2–2.2)	2.2 (1.7–2.8)	2.7 (2.5–3.0)	3.4 (3.1–3.6)	3.3 (2.7–3.9)	4.2 (3.5–4.8)	0.001	0.001
Semi-skimmed dairy products	2.8 (2.3–3.4)	2.5 (1.9–3.1)	2.9 (2.6–3.1)	2.8 (2.6–3.1)	2.0 (1.4–2.6)	2.7 (2.1–3.4)	0.051	0.574
Full-fat dairy products	4.7 (4.3–5.2)	5.0 (4.6–5.5)	4.3 (4.1–4.5)	4.4 (4.2–4.6)	4.6 (4.1–5.2)	3.9 (3.3–4.5)	0.166	0.008
Supplemented dairy products	1.1 (0.7–1.6)	2.1 (1.6–2.7)	1.4 (1.2–1.7)	2.1 (1.6–2.7)	1.0 (0.6–1.5)	1.6 (1.1–2.2)	0.214	0.476
Butter	1.8 (1.4–2.2)	1.7 (1.3–2.1)	1.6 (1.4–1.8)	1.4 (1.2–1.5)	1.4 (1.0–1.8)	0.9 (0.6–1.3)	0.458	0.029
Red meat	2.6 (2.2–2.8)	2.3 (2.0–2.5)	2.3 (2.2–2.4)	1.9 (1.8–2.0)	2.0 (1.7–2.3)	1.6 (1.3–1.8)	0.032	0.001
Low-fat processed cold meat	2.8 (2.4–3.3)	2.2 (1.8–2.7)	2.7 (2.5–2.9)	2.4 (2.2–2.5)	2.8 (2.3–3.2)	2.3 (1.8–2.8)	0.881	0.890
Processed red meat	1.1 (0.9–1.4)	0.7 (0.5–0.8)	0.7 (0.7–0.8)	0.5 (0.4–0.5)	0.5 (0.4–0.7)	0.3 (0.1–0.4)	0.001	0.001
Poultry, turkey or rabbit	3.2 (2.9–3.5)	2.8 (2.6–3.1)	3.0 (2.8–3.1)	2.8 (2.7–2.9)	2.8 (2.5–3.1)	2.8 (2.5–3.1)	0.119	0.955
Commercial sauces	1.1 (0.9–1.4)	0.5 (0.3–0.7)	0.7 (0.6–0.8)	0.4 (0.3–0.4)	0.8 (0.5–1.1)	0.3 (0.1–0.5)	0.008	0.250
Juices and/or sweetened drinks	4.7 (4.3–5.1)	4.8 (4.4–5.1)	3.3 (3.3–3.8)	2.8 (2.6–3.0)	3.1 (2.5–3.6)	0.8 (0.4–1.2)	0.001	0.001
Pastries and biscuits	4.6 (4.1–5.0)	4.6 (4.2–5.0)	4.8 (4.6–5.0)	4.2 (3.9–4.4)	4.3 (3.8–4.8)	3.3 (2.7–3.8)	0.133	0.001
Coffees	6.0 (5.0–7.1)	4.2 (3.4–5.0)	7.1 (6.5–7.7)	4.1 (3.7–4.5)	7.1 (5.9–8.4)	4.6 (3.7–5.5)	0.282	0.627
Homemade sauces	2 (0–3)	1.4 (1.1–1.7)	2 (1–3)	2.6 (2.4–2.7)	50 (5.9)	2.8 (2.4–3.3)	0.001	0.001
EVOO mL/day	20 (17–23)	15 (13–17)	29 (27–30)	30 (28–32)	33 (28–39)	44 (40–49)	0.001	0.001
Nutrition Score	−1.5 (−2.0–−1.0)	−1.4 (−1.9–−0.9)	0.5 (0.3–0.7)	2.8 (2.6–3.1)	1.8 (1.1–2.4)	6.7 (6.1–7.2)	0.001	0.0001
MEDAS Score	3.9 (3.7–4.2)	4.5 (4.3–4.7)	5.0 (4.8–5.1)	6.7 (6.6–6.9)	5.7 (5.4–6.1)	9.0 (8.7–9.2)	0.001	0.0001
Physical activity Score ≥ 0	16 (11.8)	11 (11–2)	71 (11.4)	41 (6.6)	12 (10.4)	6 (4.7)	0.845	0.127

Results expressed as Means (95% CI) and *n* (%). *p*
*^a^* Differences of mean food intakes between groups at baseline and *p*
*^b^* Differences of mean food intakes between groups at 24–28 GWs, both analyzed with the Kruskall–Wallis test. N.A., not applicable. EVOO: extra-virgin olive oil; Mediterranean Diet Adherence Screener Score. Physical Activity Score > 0: a. Walking daily (>5 days/week). Score 0: At least 30 min; Score +1, if > 60 min; Score −1, if <30 min. b. Climbing stairs (floors/day, >5 days a week): Score 0, Between 4 and 16; Score +1, >16; Score −1: <4). Targets: vegetables > 12 servings/week); fruits: > 12 servings/week; nuts > 3 servings/week; EVOO > 6 servings/week; oily fish > 3 servings/week; canned fish; wholegrain bread and cereals > 6 servings/week; white rice, bread and/or pasta < 1 serving/week; legumes > 2 servings/week; skimmed dairy products > 6 servings/week; red meat < 2 servings/week; low-fat cold meats < 2 servings/week; processed red meats < 2 servings/week; commercial sauces < 2 servings/week; juices and/or sweetened drinks < 2/week; pastries/biscuits < 2 servings/week; homemade sauces, <2 servings/week commercial sauces and >5 servings/week of sofrito; EVOO mL/day, 40 mL.

**Table 5 nutrients-11-00066-t005:** Association between late first-trimester degrees of adherence to the six food targets and adverse outcomes.

	Groups			
	Low Adherence (*n* = 136/15.6%)	Moderate Adherence (*n* = 623/71.3%)	High Adherence (*n* = 115/13.1%)	*p* Trend
GDM n (%)	41 (30.1)	121 (19.4)	15 (13.0)	0.001
Crude OR (95% CI)	1	0.56 (0.37–0.85)	0.35 (0.18–0.67)	0.003
*p*		0.006	0.002	
Insulin-treated GDM	12 (29.3)	32 (26.4)	3 (20.0)	0.512
Crude OR (95% CI)	1	0.55 (0.13–2.39)	N.A.	
*p*		0.422		
Pregnancy-induced hypertension	7 (5.1)	23 (3.7)	2 (1.7)	0.155
Crude OR (95% CI)	1	0.71 (0.30–1.68)	0.33 (0.07–1.60)	0.376
*p*		0.432	0.168	
Urinary tract infection	26 (19.1)	53 (8.5)	5 (4.3)	0.0001
Crude OR (95% CI)	1	0.39 (0.24–0.66)	0.19 (0.07–0.52)	0.001
*p*		0.0001	0.001	
Emergency-CS	8 (5.9)	30 (4.8)	2 (1.7)	0.128
Crude OR (95% CI)	1	0.81 (0.36–1.81)	0.28 (0.06–1.36)	0.287
*p*		0.606	0.115	
Perineal trauma	15 (11.0)	58 (9.3)	9 (7.8)	0.383
Crude OR (95% CI)	1	0.83 (0.45–1.51)	0.69 (0.29–1.63)	0.683
*p*		0.538	0.392	
Prematurity (<37 GW)	9 (6.6)	13 (2.1)	0 (0)	0.001
Crude OR (95% CI)	1	0.30 (0.13–0.72)	N.A.	0.026
*p*		0.007		
LGA > 90 centile	5 (3.7)	17 (2.7)	0 (0.0)	0.071
Crude OR (95% CI)	1	0.74 (0.27–2.03)	N.A.	
*p*		0.552		
SGA < 10 centile	11 (8.1)	19 (3.0)	0 (0)	0.0001
Crude OR (95% CI)	1	0.36 (0.17–0.77)	N.A.	
*p*		0.009		
NICU/observation	5 (3.7)	16 (2.6)	1 (0.9)	0.162
Crude OR (95% CI)	1	0.69 (0.25–1.92)	0.23 (0.03–1.99)	0.399
*p*		0.478	0.182	
Hyperbilurrubinemia	11 (8.1)	38 (6.1)	4 (3.5)	0.130
Crude OR (95% CI)	1	0.74 (0.37–1.48)	0.41 (0.13–1.32)	0.323
*p*		0.394	0.136	
CMFC	40 (29.4)	113 (18.1)	10 (8.7)	0.0001
Crude OR (95% CI)	1	0.53 (0.35–0.81)	0.23 (0.11–0.48)	0.0001
*p*		0.003	0.001	

Data are *n* (%) and odds ratios 95% CI (OR). *p*-trend, differences between the groups analyzed, with the χ^2^ test for the linear trend. P, unadjusted logistic regression analysis comparing ORs between different degrees of adherence, as compared to the reference group (low adherence group). High adherence: 5–6 targets; moderate adherence 2–4 targets; low adherence 0–1 targets (group of reference). CI, confidence interval; GDM, gestational diabetes mellitus; CS, cesarean section; LGA, large-for-gestational-age; SGA, small-for-gestational-age; NICU, neonatal intensive care unit; CMFC, composite of materno-foetal complications; CMFC: having at least one of: emergency C-section, perineal trauma, pregnancy-induced hypertension and preeclampsia, prematurity, large-for-gestational-age and small-for gestational age.

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
