# Peer review of "A High Adherence to Six Food Targets of the Mediterranean Diet in the Late First Trimester is Associated with a Reduction in the Risk of Materno-Foetal Outcomes: The St. Carlos Gestational Diabetes Mellitus Prevention Study"

_nutrients, 2018, doi:10.3390/nu11010066_

Round 1

Reviewer 1 Report

The manuscript by Assaf-Balut and colleagues describes a post-hoc analysis of the St. Carlos gestational diabetes mellitus (GDM) Prevention Study, which aims to assess the effect of late first-trimester adherence to a Mediterranean Diet pattern on maternal and foetal complications. In my opinion, the main topic is of interest, since there is the need to uncover what are the main risk factors of adverse pregnancy outcomes, using an approach based on dietary pattern analysis. However, the manuscript has several weaknesses.

First of all, I extremely recommend an extensive editing of English style and language. There are several errors, typos and confusing paragraphs throughout the text.

My major concern is about dietary assessment. As I understood, the Authors used two semi-quantitative questionnaires: the Diabetes Nutrition and Complications Trial (DNCT) and MEDAS questionnaires. The DNCT questionnaire included 15 items, out of which 12 food frequency intakes (daily or weekly). The second one was an adapted version of the MEDAS questionnaire, a 14-item questionnaire developed and validated in the PREDIMED study. Particularly, the authors did not account for the consumption of alcohol or juice because both are misadvised during pregnancy. They also accounted for vegetable intake regardless of being raw or cooked. Thus, the ideal MEDAS score was considered >8, with slightly different criteria than in the MEDAS questionnaire. Moreover, the six food group targets used did not include whole-wheat cereals, pulses and fish.

Given these limitations, the Authors should better explain the reasons why they made several adaption to the MEDAS criteria, what is the generalizability to the Mediterranean Diet and how did they merge data from DCNT and MEDAS questionnaire.

With regard statistical analyses, I suggest to remove the ROC curve analysis, since it usually illustrates the diagnostic ability of a test. In the context of the present manuscript, it is meaningless.

In the result section, the Authors used the term “rate” instead of incidence.

Author Response

Reviewer 1.

Thank you very much for your kind comments and constructive suggestions. We agree with you on most points. We have made the corresponding changes to improve the article.

Please note that we have changed “six food group targets” to “six food targets”. While it is true that we use six food targets, it only includes five food groups (for extra virgin olive oil we set two different targets). This had led to minor changes in the title, figure 1 and the rest of the manuscript. Due to changes suggested by reviewer 2, the order of the first 12 references has changed (lines 476-553).

The changes applied are as follows:

The manuscript by Assaf-Balut and colleagues describes a post-hoc analysis of the St. Carlos gestational diabetes mellitus (GDM) Prevention Study, which aims to assess the effect of late first-trimester adherence to a Mediterranean Diet pattern on maternal and foetal complications. In my opinion, the main topic is of interest, since there is the need to uncover what are the main risk factors of adverse pregnancy outcomes, using an approach based on dietary pattern analysis. However, the manuscript has several weaknesses.

First of all, I extremely recommend an extensive editing of English style and language. There are several errors, typos and confusing paragraphs throughout the text.

In agreement, a native English-speaking colleague has revised this manuscript.

My major concern is about dietary assessment. As I understood, the Authors used two semi-quantitative questionnaires: the Diabetes Nutrition and Complications Trial (DNCT) and MEDAS questionnaires. The DNCT questionnaire included 15 items, out of which 12 food frequency intakes (daily or weekly). The second one was an adapted version of the MEDAS questionnaire, a 14-item questionnaire developed and validated in the PREDIMED study. Particularly, the authors did not account for the consumption of alcohol or juice because both are misadvised during pregnancy. They also accounted for vegetable intake regardless of being raw or cooked. Thus, the ideal MEDAS score was considered >8, with slightly different criteria than in the MEDAS questionnaire. Moreover, the six food group targets used did not include whole-wheat cereals, pulses and fish.

Given these limitations, the Authors should better explain the reasons why they made several adaption to the MEDAS criteria, what is the generalizability to the Mediterranean Diet and how did they merge data from DCNT and MEDAS questionnaire.

There are two different aspects. One is the slightly different criteria used in the MEDAS questionnaire. The other is the criteria used to determine the “six food targets”.

1.        The MEDAS questionnaire was adapted for pregnant women. This translated in two main changes. One, not accounting for alcohol consumption since it is contraindicated during pregnancy. And two, not considering intake of fresh juice the same as intake of a piece of fruit. The consumption of juice (fresh or bottled) has been associated with increased risk of developing GDM. Therefore, it is misadvised during pregnancy (Ramos-Leví et al. Int. J. Endocrinol. 2012, 2012, 1–9, doi:10.1155/2012/312529; and Ruiz-Gracia, T et al. Clin. Nutr. 2016, 35, 699–705, doi:10.1016/j.clnu.2015.04.017.  An additional minor change was that we did not limit vegetable consumption to having at least one serving in raw or as a salad. Pregnancy guidelines indicate that vegetables should be cleaned throughly if they are to be eaten raw. Some women limit their intake of raw vegetables during pregnany, and prefer to eat cooked vegetables.  If we were to evaluate at least one serving/day of raw vegetables we could be underestimating overall vegetable consumption. The several adaptions made to the MEDAS criteria are explained in the manuscript (lines 142-158).

2.        The six food targets included four of the most important elements of the MedDiet: EVOO (weekly and daily consumption), nuts, vegetables and fruits (not as fruit juices). Moreover, EVOO and nuts are tools usually used to ensure compliance to the MedDiet (Estruch, R et al. N. Engl. J. Med. 2013, 368, 1279–1290, doi:10.1056/NEJMoa1200303). The modifications made to the MEDAS questionnaire and score were perfomed to provide a questionnaire that can be adapted to pregancy, as explained above.  In addition, our results show that the higher the adherence to the six food targets, the higher the MEDAS score. In fact, women who complied with the 6 food targets had the highest consumption of vegetables, fruits, nuts, EVOO (days/week), oily fish, whole-grain cereals, legumes, skimmed dairy products, EVOO (ml/day) and homemade sauces. This was also in parallel to a lower consumption of refined flours, red and processed meat, juices/sweetened drinks, pastries and biscuits. Overall, these women had a MEDAS score of 9. Women in this group had lower risks of having complications such as GDM, urinary tract infections, prematurity, small-for-gestational-age new-borns and a composite of maternofoetal complications. This seems to indicate that using this adapted MEDAS questionnaire in pregnancy could be appropriate.

3.        With regards to merging the DNCT and MEDAS questionnaire, I think this was misinterpreted. The DNCT and MEDAS questionnaires were not merged. They are two independent questionnaires that we used to evaluate different dietary patterns. The DNCT questionnaire evaluates general healthy eating habits and was used to calculate the “Nutrition Score”. On the other hand, the MEDAS questionnaire was used to evaluate adherence to the MedDiet specifically. It was used to calculate the “MEDAS score”. To assess the adherence to the six food targets we used information provided by the MEDAS questionnaire.

With regard statistical analyses, I suggest to remove the ROC curve analysis, since it usually illustrates the diagnostic ability of a test. In the context of the present manuscript, it is meaningless.

While the ROC curve analysis is used to evaluate sensitivity and specificity of a test, it can also be used to analyse the predictive value of the selected risk factors (6 food targets) and the development of GDM and maternofoetal complications. We believe that it does provide important information. However, we also agree that it might not be necessary to include in the main text (as also suggested by Reviewer 2). Therefore, we have included Figure 3 as supplementary material (now Supplemtary Figure S1. a and b.).

In the result section, the Authors used the term “rate” instead of incidence.

This has been changed. 

Reviewer 2 Report

The paper presented by Assaf-Balut et. al. analyzed late first-trimester dietary habits of women that complied with the study protocol of the St. Carlos GDM Prevention study. The analysis was to assess the effect of late first-trimester degree of adherence to a MedDiet pattern, using six items, on materno-fetal complications.

The subject taken by the authors is very interesting because the prenatal period is a critical moment in the development of diseases in the mother and the newborn.

But after careful analysis of the manuscript, I have a few general comments:

Line 34: there should be 2-4 targets

Introduction:

The introduction can be improved by giving more background of this specific groups (pregnancy)  with their current issues of nutrition. An overview of existing (current) research requires some improvement.

Line 48-51 is not clear, make sure it is a proper reference to the literature.

Also, make sure that some fragments should not be into the Materials and Methods section e.g. line 72-77

Materials and Methods.

The section Materials and Methods requires ordering and strengthening, e.g. unclear division into groups, anthropometric measurements  (please list out the equipment used to measure heights and weights: company, model number, etc; measurement conditions), blood analysis (HbA1C, Cholesterol, Triglycerides TSH ...), and blood pressure - table 1?

Also, (please list out the equipment used to measure heights and weights: company, model number, etc).

Results

Consider the order in which you discuss the results because, in my opinion, tables 3 and 4 should be swapped.

Overall the tables are difficult to read and lack a uniform mode of data presentation. They would benefit from a clearer explanation of which values were compared where significant differences were detected.

Provide all the explanations used in the table. The table should stand alone.

Discussion

Make sure all of the discussed incidences are properly discussed and explained.

 In my opinion, figure 3 can be moved to the supplementary materials.

Tables in the supplementary should be in English,  furthermore, it should be considered whether they are necessary?

A minor concern is that there a numerous grammar and spelling errors.

Author Response

Thank you very much for your kind comments and constructive suggestions. We agree with you on most points. We have made the corresponding changes to improve the article.

Please note that we have changed “six food group targets” to “six food targets”. While it is true that we use six food targets, it only includes five food groups (for extra virgin olive oil we set two different targets). This had led to minor changes in the title, figure 1 and the rest of the manuscript.

The changes applied are as follows:

Comments and Suggestions for Authors

The paper presented by Assaf-Balut et. al. analyzed late first-trimester dietary habits of women that complied with the study protocol of the St. Carlos GDM Prevention study. The analysis was to assess the effect of late first-trimester degree of adherence to a MedDiet pattern, using six items, on materno-fetal complications.

The subject taken by the authors is very interesting because the prenatal period is a critical moment in the development of diseases in the mother and the newborn.

But after careful analysis of the manuscript, I have a few general comments:

Line 34: there should be 2-4 targets

As suggested, this has been amended.

Introduction:

The introduction can be improved by giving more background of this specific groups (pregnancy) with their current issues of nutrition. An overview of existing (current) research requires some improvement.

Following this suggestion, we have tried to improve the introduction giving more detail regarding pregnancy and nutrition (62-66).

Line 48-51 is not clear, make sure it is a proper reference to the literature.

In agreement to this comment, we have added references to this line (1-10). This has led to changes in the order of first 12 references, as shown from lines 483-560.

Also, make sure that some fragments should not be into the Materials and Methods section e.g. line 72-77

We included this section here because it outlines what six food targets are going to be used in the study and why.  This is important for readers to be able to comprehend the aim of the study. While a more detailed description of this fragment is provided in the Materials and methods section, we believe it is important to leave this summarized fragment here to better understand the objective of the study.

Materials and Methods.

The section Materials and Methods requires ordering and strengthening, e.g. unclear division into groups, anthropometric measurements  (please list out the equipment used to measure heights and weights: company, model number, etc; measurement conditions), blood analysis (HbA1C, Cholesterol, Triglycerides TSH ...), and blood pressure - table 1?

Also, (please list out the equipment used to measure heights and weights: company, model number, etc).

As suggested, we have added the requested information to this section (lines189-193 and 203-219).

Results

Consider the order in which you discuss the results because, in my opinion, tables 3 and 4 should be swapped.

As requested, former table 4 is table 3 (lines 266-276) and former table 3 is now table 4 (lines 277-296). This lead to having to add a new sub-section labelled “3.4. Dietary habits according to late first-trimester degree of adherence to six food targets at baseline and at 24-28 GWs” (lines 277-278). Consequently, former point 3.4 is now point 3.5 (line 307).

Overall the tables are difficult to read and lack a uniform mode of data presentation. They would benefit from a clearer explanation of which values were compared where significant differences were detected.

Provide all the explanations used in the table. The table should stand alone

In agreement to this suggestion, we have provided a clearer explanation of the mode of data presentation in all tables. We have added to the footnotes of all tables a brief phrase explaining what variables the p-value is comparing in each case and the statistical analysis used:

Tables 1 and 4 (former) present results as mean ±SD or n (%) (lines 274-275). We believe it is the most appropriate way to present descriptive information. We have added “n(%)” to these tables.

Table 2 and 3(former) show data as mean (95% CI) or n (%) (lines 253-254 and 288-289, respectively).We believe it is the most appropriate way to present data of variables that are related.

Table 5 shows results as n(%), not as mean ±SD (lines 316-319). We have amended this error.

Figure 2: lines 351-353.

Discussion

Make sure all of the discussed incidences are properly discussed and explained.

We believe that in the discussions section we have discussed all results. Results from Table 2 and former table 4  have been discussed in lines 381-400; table 3, 5 and figure 2 in lines 401-439. However, we have elaborated more on results from Table 4 (former Table 3) (lines 392-400). In addition, this paragraph has been moved from lines 377-380 to lines 397-400.

Nevertheless, if you think we should elaborate further on some results or on specific aspects we are willing to make further amendments.

 In my opinion, figure 3 can be moved to the supplementary materials.

In agreement with this reviewer, we have included Figure 3 as supplementary material (now Supplementary Figure S1.a. and b.

Tables in the supplementary should be in English, furthermore, it should be considered whether they are necessary?

After considering this suggestion, we have decided not to include this dataset. We have inserted a “Data statement” informing that “the datasets generated during and/or analysed during the current study are available from the corresponding author on reasonable request” (lines 460-461).

A minor concern is that there a numerous grammar and spelling errors.

A native English-speaking colleague has revised grammar and spelling errors..

Round 2

Reviewer 1 Report

The Authors accomplished all my comments and suggestions, improving the quality of their manuscript. I only recommend to remove ROC curve analysis or to strongly motivate the reason to maintain it. 

Author Response

Reviewer 1.

The Authors accomplished all my comments and suggestions, improving the quality of their manuscript. I only recommend to remove ROC curve analysis or to strongly motivate the reason to maintain it.

Thank you for your suggestion. After careful consideration, we have decided to remove ROC curve analysis. Consequently, we have removed any reference to it in the sections of Materials and methods (lines 216-217) and results (309-313). 

Reviewer 2 Report

Thank you for addressing my comments in the revised version of the manuscript. Overall, the manuscript has been strengthened as a result.

However, the tables should be improved  (e.g. add column titles everywhere) - adapting to the editorial requirements, for more clarity.

Author Response

Thank you for addressing my comments in the revised version of the manuscript. Overall, the manuscript has been strengthened as a result.

However, the tables should be improved  (e.g. add column titles everywhere) - adapting to the editorial requirements, for more clarity.

Thank you for your suggestions. We have changed all tables, trying to adapt them to editorial requirements. We hope to have made the expected modifications.

Please note that following Reviewer 1’s advice, we have removed the ROC curve analysis. Consequently, we have removed any reference to it in the sections of Materials and methods (lines 216-217) and results (309-313).
